# The role of gestures in autobiographical memory

**Cagla Aydin**[1]*, **Tilbe Göksun**[2], **Ege Otenen**[1], **Selma Berfin Tanis**[1], **Yağmur Damla Şentürk**[1]

**1** Department of Psychology, Sabancı University, Istanbul, Turkey, **2** Department of Psychology, Koç University, Istanbul, Turkey

* cagla.aydin@sabanciuniv.edu

**Data Availability Statement:** The data underlying the results presented in the study are available from Open Science Framework at https://osf.io/q42p3/.

## Abstract

Speakers employ co-speech gestures when thinking and speaking; however, gesture's role in autobiographical episodic representations is not known. Based on the gesture-for-conceptualization framework, we propose that gestures, particularly representational ones, support episodic event representations by activating existing episodic elements and causing new ones to be formed in the autobiographical recollections. These gestures may also undertake information-chunking roles to allow for further processing during remembering, such as a sense of recollective experience. Participants ($N = 41$) verbally narrated three events (a past autobiographical, a future autobiographical, and a non-autobiographical event) and then rated their phenomenological characteristics. We found that, even though gesture use was not different across the three event conditions, representational gestures were positively associated with the episodic event details as well as their recollective quality within the past autobiographical event narratives. These associations were not observed in future event narrations. These findings suggest that gestures are potentially instrumental in the retrieval of details in autobiographical memories.

## Introduction

Abundant research has suggested that gestures are not just used in the service of communication (e.g., [1]) but are also involved in the speakers' cognitive processes and mental representations (e.g., [2]). Autobiographical memories are complex mental representations of the self that are constructed from different types of episodic and semantic knowledge, and thought to contain sensorimotor elements [3]. Provided that autobiographical memories are not fully amodal representations, here, we seek a link between gesture production and autobiographical memory representations. This idea complements well with the embodied cognition literature, where memory retrieval has been shown to benefit from sensorimotor activations during encoding (e.g., [4]). It has been shown that embodied forms of remembering involve different modalities, including eye-movements (e.g., [5]), body postures (e.g., [6]), and movements (e.g., [7]). Similarly, the beneficial role of gesture production has been shown during episodic memory (e.g., [8, 9]) and autobiographical memory retrieval [10]. However, to our knowledge, no previous studies attempted to explore the possible contribution of gesture production

**Funding:** The author(s) received no specific funding for this work.

**Competing interests:** The authors have declared that no competing interests exist.

during the retrieval of complex autobiographical representations in adults. Given this backdrop (e.g., [6, 11], for a review), we aim to examine the link between gesture production and autobiographical memory representations.

Co-speech gestures are intertwined with speech, temporally and semantically [12]. Gestures are mainly considered in four categories: *iconic gestures* depict action, motion, and shape or indicate location and trajectory; and *metaphoric gestures* represent abstract ideas. *Beat gestures*, on the other hand, are simple motoric movements produced along with the rhythm of the speech. Last, *deictic gestures* include the type of hand movements that involve pointing or reaching to objects or abstract entities [12]. Recent frameworks mainly consider iconic and metaphoric gestures without deictic ones as representational and beat gestures as non-representational [1, 2]. In this paper, we follow the same categorization as we focus on specific representations of concrete and abstract concepts in iconic and metaphoric gestures (see [13, 14] for similar categorizations). Since representational gestures are visuospatial in nature [2], this categorization will allow us to investigate the possible contribution of visuospatial information to autobiographical remembering.

Recently, representational gestures have been highlighted as components that affect and even shape mental representations [11, 15]. According to the gesture-for-conceptualization framework [2], gestures may change the content of the thought by representing and then triggering and adding new sensory-spatial-motoric representations. Other roles they assume include exploring spatio-motoric information and organizing them into packages that allow an effective encoding [2, 16]. Taken together, these functions have great implications for how autobiographical representations would take their final form during retrieval, as autobiographical memory is thought to be "instantiating context for sensory-perceptual episodic memory" [3 (p. 1)].

What does an autobiographical event representation entail? A full representation includes episodic elements such as events, sensory-perceptual information (across different sensory modalities), actions, people, and spatial and temporal relations between them. An associated phenomenological sense of "autonoetic consciousness", which is described as a sense of time as personal events are happening in relation to the self, is considered as the hallmark of thinking about the personal past [17 (p. 1)]. In addition to these episodic elements, semantic informational components, such as general knowledge about the self and the world, are also included. Therefore, multiple processes are in effect to retrieve, choose, and bind together relevant pieces of information [18, 19]. Our position in the present study is that, when there is an opportunity, co-speech hand gestures may contribute to those processes owing to the multimodal nature of memory representations in the gesture and speech interaction of narrating episodic events. To our knowledge, this is the first study to examine the link between gestures and various forms of episodic representations.

Representational gestures may support autobiographical memory retrieval in several ways. During recollection, relevant sensorimotor and perceptual components of past experiences are reactivated [11, 20]. When some of these episodic components are captured in gesture format, based on Kita et al.'s framework [2], gestures may activate and manipulate further spatio-temporal information, such as the location or relative positioning of the objects or people. Since the construction of an autobiographical event is a flexible and dynamic process that allows the details to be bound differently at each retrieval [21], the influence of gesture production during a retrieval attempt is critical for the type and amount of event details to be included. Supporting this notion, previous research has shown that gesture use and the number of correct details (e.g., shape, location) in (non-autobiographical) event accounts are related (e.g., [22]; see also [23]). Also, Cook et al. [8] showed that when participants were encouraged to use gestures during event description, both immediate and later recall performance increased, compared to the

no-gesture group. This effect was present for events describing both complex everyday activities, which are highly visuospatial in nature, and static images that depict hand-movements. Therefore, we expect to see a positive association between the rate of representational gesture use and the number of episodic elements in an autobiographical event narrative.

Based on Kita et al. [2], representational gestures also assume a packaging function by chunking information into units that can be used in other operations. Applied to the mechanisms of autobiographical memory, packaging the sensory-perceptual details in co-speech gestures during recollection may facilitate other operations, such as recollective phenomenology; i.e., vividness and sense of reliving. Previous evidence has shown that what gives rise to recollective phenomenology, such as the sense of re-experiencing the event or vividness, may be the presence of sensory details [24]. Similarly, it has been suggested that the subjective sense of vividness may be determined by the amount of sensory information [25] and the number of retrieved memory details [26, 27]. Therefore, it is likely that as the representational gesture rate increases, the phenomenological experience of a memory would be stronger due to the availability of sensory-perceptual elements in the representation [28]. We maintain that owing to their nature involving spatio-motoric and sensory details, episodic representations would be in direct communication with the gesture system and would be open to their contribution to the episodic richness of the remembered events.

A frequent discussion in the autobiographical memory literature is the shared mechanisms of remembering the past and imagining the future events (e.g., [29, 30]). Owing to similar event construction processes, findings have shown that imagined personal events are based on the same informational and phenomenological components provided by episodic as well as semantic memory [19]. Therefore, the difference in the level of episodic content between past and future personal events is thought to be only quantitative (i.e., a matter of degree in the content details) rather than qualitative [18]. In fact, remembered events are typically shown to contain more perceptual content than imagined future events (e.g., [31, 32]). Thus, if gesture production supports detailed episodic representations, we would expect representational gesture rates to be also positively correlated with the episodic details in imagined personal events, but to a lesser degree than in remembered events.

## The present study

In the present study, we ask whether representational gestures support autobiographical episodic event representations. Evidence shows that gestures are powerful enough to change the content of thought when speaking (e.g., [16, 28]). Given that memory processes flexibly reconstruct an autobiographical event representation by binding together different perceptual and spatio-motoric details during retrieval [21], it is conceivable that gestures would contribute to this process by representing already existing and activating new episodic representations [2]. They may even help in "packaging" episodic information into functional units to be used in other processes, such as recollective phenomenology, which is a critical component of an autobiographical experience.

Acknowledging that it is highly likely that the relationship between memory and gestures is bidirectional [33], we propose that thinking about episodic autobiographical events would activate gesture use, and the use of gestures, then, would benefit the episodic richness of the event representations. In other words, based on the literature that demonstrates the benefits of gesture use during other types of remembering [22, 34], we contend that gestures would, in a sense, have a facilitative role in episodic thinking.

As an initial step in understanding the association between gesture use and episodic representations, we explored the degree of gesture use across three conditions: a past

autobiographical event condition, a future autobiographical event condition, and a non-auto-biographical event condition. The existence of having both a future event and a non-autobio-graphical event condition ensured that the differences are not due to the constructive demands of the event generation process but to the episodic nature of the events. We expected the ges-ture rate to be higher when individuals remember past personal events compared to when they imagine future personal events. Previous evidence has shown that speakers gesture at a higher rate when they have specific motor experience with the information they are describing com-pared to when they do not [35]. We also expect the gesture rate to be lower in the non-autobio-graphical event condition when compared to past and future personal events because the former would lack episodic quality based on a recent finding that individuals gestured less when they described procedural events vs. emotional experiences [36].

Importantly, we also investigate whether representational gesture use is positively associ-ated with the particular episodic elements in past and future events. To quantify the episodic elements, we used a widely adopted instrument, the Autobiographical Interview (AI; [37]; see Method for the coding details). Within the narratives, details specific to the main event, such as place and perceptual information, are classified as *internal details*. These details reflect quali-ties of recollective experience and are considered as an index for how much episodic detail an event account contains; commonly referred to as *episodic specificity*. Details that are not spe-cific to the main event, including general conceptual and personal information are categorized as *external details*, in addition to repetitions and off-topic comments. These are considered as indices of semantic memory. Our question, here, is whether the number of internal details would be associated with the number of representational gestures produced. The present approach allows us to delve into the mechanisms behind the relationship between gesture use and particular detail categories among the episodic (internal) elements. For instance, a positive relationship should be observed between place details and representational gestures if gestures do facilitate the maintenance of spatial representations or highlight them to be used in further processing, as previously suggested (e.g., [9]).

Although the main question of this study is about the role of representational gestures on episodic remembering, due to the exploratory nature of the present study, we also examine how non-representational gestures would be associated with the memory details. Even though there is less consensus on the functional role of non-representational gestures, they are fre-quently discussed within the discourse-fluency contexts [38, 39], we tentatively expect them to be correlated with the external details as these details pave the way to the actual specific event, and they help contextualizing the episodic components.

## Method

### Participants

Forty-four undergraduate students participated in return for course credit. Data from three participants were excluded due to hand-arm injuries (2) and recording error (1). The final sample consisted of 41 participants ($M_{age}$ = 19.5, $SD_{age}$ = 0.7; 60% female). The sample size is determined based on the previous studies in the literature (e.g., [8, 40–42]). Additionally, post-hoc power analysis suggests that our sample of 41 people successfully detects the power of .85, with a false positive rate of .05 under the estimated effect size of the study ($d$ = 0.44).

All participants were native Turkish speakers. Participants were recruited through the sub-ject pool of Sabancı University. The study was approved by the Institutional Review Board of Sabancı University (FASS-2018-17).

## Materials and procedure

Experimental sessions took place in the laboratory. Participants sat on an armless chair across the experimenter (no table in between) to allow for spontaneous gesture use when speaking; however, gesture use was not particularly mentioned. They were reminded to communicate as naturally as possible. They would be talking about everyday conversation type of topics.

Participants verbally narrated three different types of events (past autobiographical, future autobiographical, and non-autobiographical) in response to prompts presented to them. The non-autobiographical events were probed by asking the participants to narrate a typical instance of a certain event (i.e., a typical bank transaction procedure and a typical flag-raising morning ceremony in school, which is a common practice in some countries where students gather in the schoolyard to sing the national anthem together). Cue words that were spatial in nature (e.g., kitchen, street) were used to sample past autobiographical events and future auto-biographical events. The order of the cue words within each condition was counterbalanced each time. The participants received the instructions for the non-autobiographical event first, and the order of past and future events was also counterbalanced across participants. They were particularly asked to recall past autobiographical events and imagine possible future events that are precise and specific in time and place, which last more than a few minutes but less than 24 hours (adapted from [43, 44]). In the non-autobiographical event condition, they were asked to narrate a procedural event (i.e., a typical bank transaction) to somebody who does not know anything about this world's affairs. The sessions were videotaped for later coding of event details and gestures.

After describing each of the past and future autobiographical events, participants rated their subjective experience about that event. These items were selected from the Memory Characteristics Questionnaire (MCQ; [45]) and the Autobiographical Memory Questionnaire (AMQ; [46, 47]). For the purposes of the present study, only the items that measured vividness (from MCQ; [45]), reliving, and mental time travel (from AMQ; [46]; also, [47]) were included. For instance, a typical item for measuring the sense of reliving would ask the participants to rate the following statement on a 5-point Likert scale: "While remembering the event, I feel as if I am reliving it." Previously it was shown that, under some conditions, gestures may increase the cognitive load (e.g., [48]) and frequent gesturing was associated with lower scores in working memory (e.g., [49, 50]). Therefore, we also measured the visuospatial working memory by administering the Corsi Block-Tapping Task (CBT; [51]; computerized administration) along with the Mental Rotation Test (MRT; [52]; paper-pencil administration) to measure spatial imagery ability. The order of these tasks was counterbalanced across participants.

## Event narrative coding

Memory narratives were transcribed verbatim from the video recordings with all utterances including false starts, fillers, and other disfluencies in addition to words. Then, episodic details in the past and future memory narratives were coded according to the coding scheme of the Autobiographical Interview (AI; [37]) by two independent researchers who were blind to the objectives of the study. Non-autobiographical events were not coded, as by nature they are not episodic events. Following the coding instructions in Levine et al. [37], for each narrative, the main event was initially identified. Then, all the unique details about that event were coded as internal details while all the other information was coded as external details. Information related to the main event (internal details) was categorized into *event*, *time*, *place*, *emotion/thought*, and *perceptual* details. To exemplify: "When I met with my friend at the coffee shop, I realized that I missed her very much." consists of two event details (meeting, with a friend), one place (coffee shop), and one emotion/thought (the feeling of missing) details. External

details consist of the details which are about other events unrelated/secondary to the main event, *semantic* details which are the general information related to the self or the world, *repetitions*, and other non-episodic details which do not fit any of the detail categories. Semantic details were further classified into *General Semantic* (GS; e.g., "As you know, the instant coffee has less caffeine than regular coffee") and *Personal Semantic* details (*Self-Knowledge* [SK], e.g., "I hate coffee"; *Autobiographical Facts* [AF], e.g., "I was born in 1997"; *Repeated Events* [RE], e.g., "I go to school every day") according to Renoult et al.'s [53] coding scheme. Therefore, non-episodic general knowledge about the world and the self was coded with separate detail categories.

After the coding, each piece of unique information was scored as 1 in a given detail category. Internal and external detail scores were then calculated by summing their relevant memory details. The inter-rater reliability was calculated by using 25% of the data. For internal details, the intraclass correlation coefficient (ICC; one-way random effect model, [54]) indicated excellent agreement with .99 and good agreement for external details with .88 ICC score.

### Gesture coding

Participants' spontaneous co-speech gestures were coded for each representative (i.e., referring to object shapes, object sizes, actions, directions of movement, and abstract concepts such as time) or rhythmic hand movement (not including self-adaptors) from the video recordings. Two independent researchers manually coded the video recordings using the ELAN software package (version 6.2, [55]). The coders first detected if any gesture was produced in a motion, then classified it under one of the gesture categories. These four categories were: iconic, deictic, beat [12], and other gestures -which cannot be classified under any of the aforementioned categories-. Interrater reliability was computed by two independent raters coding 20% of the dataset. Cohen's K [56] was computed to be .796 (*SD* = 0.035) which indicates moderate-to-strong agreement between the coders. The analyses for the present study were conducted by using iconic (as representational) and beat (as non-representational) gestures. Other types of gestures (i.e., deictic, *M* = .63, *SD* = 2.56; other, *M* = .20, *SD* = 1.31) were low in frequency, possibly due to the nature of the events being described.

## Results

### Analytic plan

To examine gesture rate differences across conditions, we run a repeated measures ANOVA with the type of event (past, future, non-autobiographical) as the within-subjects factor. Then, we focus on the relationship between the individual differences (visuospatial and spatial imagery skills) and gesture rates with a Kendall's tau rank correlation analysis. In line with the present study's exploratory nature, we ran a series of Kendall's tau rank correlation analyses in order to see the relations between gesture use and internal and external memory details. First, we test the association between internal details and representational gestures in the past, then in the future events. Next, we explore external details and their relationship with non-representational and representational gestures in past and future events, respectively. Fisher's Z-transformation is used to compare correlation coefficients between conditions. As the final step, we explore the relationship between representational gesture use and memory phenomenology ratings (vividness, reliving, mental time travel).

In order to see whether there were any differences in terms of gesture rate across conditions, we calculated gesture per utterance (i.e., gesture use with respect to all verbal utterances) by dividing the total number of gestures (i.e., summation of representational, non-representational, deictic, and other gestures) for each event narrative by the total number of utterances

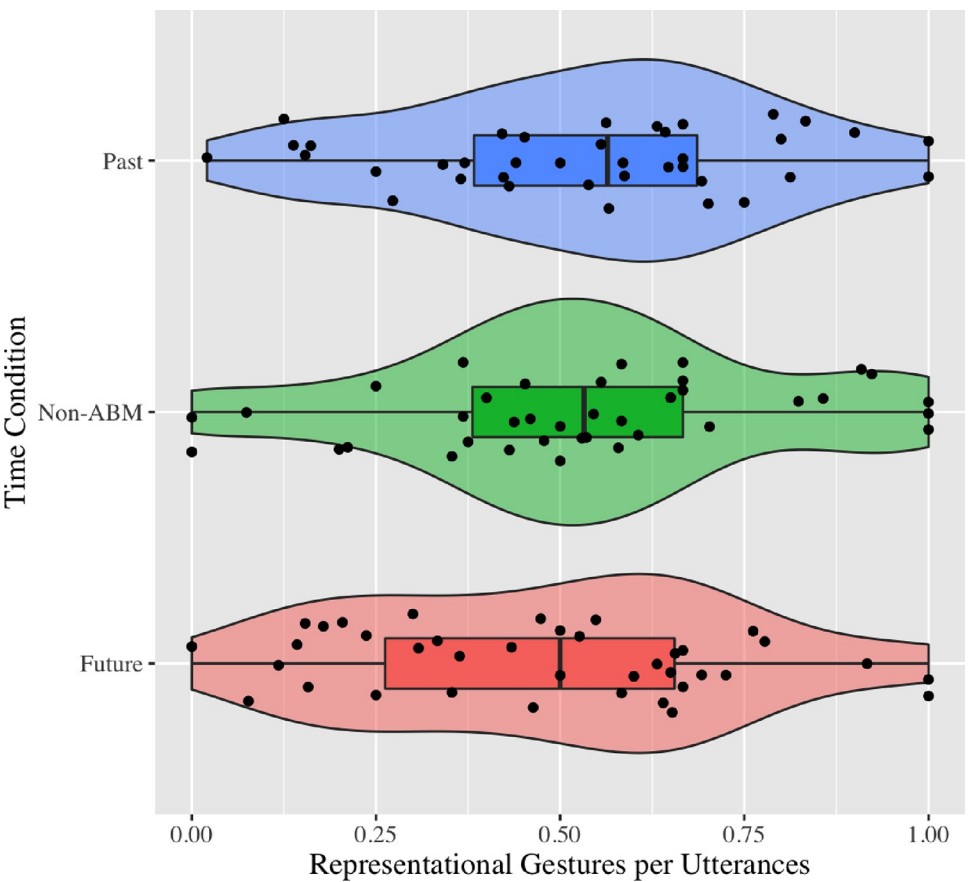

**Fig 1. Scatterplots illustrating relations between event conditions and representational gesture rate.** *Note*. None-ABM corresponds to non-autobiographical memories.

(see Fig 1, for representational gestures). This analysis revealed that past, future, and non-autobiographical events differed from each other $F(2, 80) = 3.44$, $p = .04$, $\eta2 = .02$, yet the differences could not survive the Tukey's test for post hoc analysis.

When we controlled for the influence of visuospatial skills and spatial imagery skills through participants' CBT and MRT scores, we did not observe any relation of these skills on gesture rate in any condition (past: CBT, $\tau = .06$, $p = .62$, MRT: $\tau = .10$, $p = .34$; future: CBT, $\tau = .00$, $p = 1.0$, MRT $\tau = .06$, $p = .60$); non-ABM: CBT, $\tau = -.04$, $p = .76$: MRT, $\tau = .03$, $p = .77$; for CBT and MRT, respectively). Although we did not observe the effect of individual differences measured through these metrics, we have found that representational ($\tau = .34$, $p = .004$) gestures were correlated in future and past conditions as well as non-ABM and past conditions ($\tau = .23$, $p = .05$). Past and future conditions were also correlated ($\tau = .38$, $p = .001$) in terms of the non-representational gesture rate. These correlations may reflect a gesturing tendency at the individual level regardless of the content, yet the measures in the current study were not able to detect it.

Table 1. presents the total number of gestures, means, and standard deviations broken down by event detail categories.

The main question for the present study was: Are gesture types related to the types of event details? We started out with the most critical question by examining whether representational gestures were associated with internal event details. To be consistent with the literature using

**Table 1. Means and standard deviations.**

| | Past | | Future | | Non-autobiographical | |
|---|---|---|---|---|---|---|
| | **Total** | ***M (SD)*** | **Total** | ***M (SD)*** | **Total** | ***M (SD)*** |
| Representational gestures | 531 | 13 (12.8) | 340 | 8.29 (7.32) | 372 | 9.07 (6.69) |
| Non-representational gestures | 476 | 11.6 (11.7) | 395 | 9.63 (9.65) | 328 | 8 (7.79) |
| Internal details | 674 | 16.4 (16.6) | 505 | 12.3 (11) | - | - |
| Event | 439 | 10.7 (11.2) | 296 | 7.22 (7.92) | - | - |
| Time | 27 | 0.66 (0.76) | 24 | 0.59 (0.95) | - | - |
| Place | 47 | 1.15 (1.30) | 49 | 1.20 (1.33) | - | - |
| Perceptual | 104 | 2.54 (3.57) | 98 | 2.39 (3.46) | - | - |
| Thought/Emotion | 57 | 1.39 (3.35) | 38 | 0.93 (1.98) | - | - |
| External details | 577 | 14.1 (10.9) | 386 | 9.41 (7.64) | - | - |
| Event | 20 | 0.49 (1.12) | 40 | 0.98 (1.90) | - | - |
| Time | 3 | 0.07 (0.26) | 2 | 0.05 (0.22) | - | - |
| Place | 2 | 0.05 (0.22) | 0 | 0 (0) | - | - |
| Perceptual | 1 | 0.02 (0.16) | 2 | 0.05 (0.31) | - | - |
| Thought/Emotion | 30 | 0.73 (1.28) | 51 | 1.24 (1.97) | - | - |
| Self-knowledge (SK) | 71 | 1.73 (2.95) | 49 | 1.20 (2.05) | - | - |
| Autobiographical facts (AF) | 141 | 3.44 (3.38) | 68 | 1.66 (3.01) | - | - |
| Repeated events (RE) | 104 | 2.54 (4.11) | 21 | 0.51 (1.00) | - | - |
| General semantics | 34 | 0.83 (1.48) | 13 | 0.32 (0.76) | - | - |
| Repetitions | 54 | 1.32 (2.13) | 29 | 0.71 (1.27) | - | - |
| Other | 117 | 2.85 (3.44) | 111 | 2.71 (3.83) | - | - |

*Note.* Representational gestures refer to iconic, and non-representational ones refer to beat gestures. Since the non-autobiographical events are not episodic in nature, they were not coded for internal and external details. All values in the Total column represent count data.

the Autobiographical Interview [37, 57], we calculated the total count scores both for internal details and for gesture use. Since the data were not normally distributed, we calculated Kendall's tau-b for the correlations between memory details and gesture use. For summaries of the relevant correlations, please refer to Tables 2 and 3.

We confirmed our predictions in the past autobiographical event condition. For the past events, representational gestures correlated with internal details ($\tau = .27$, $p = .02$). To delve into which particular episodic details were related to the representational gestures, we looked into internal detail categories. Representational gestures were correlated with internal place details ($\tau = .28$, $p = .02$) as well as internal perceptual details ($\tau = .36$, $p = .002$) as depicted in Table 2, confirming our prediction that gestures may help individuals to sustain spatial and sensorimotor information associated with the mental representations stored in the memory. Even though we did not have specific predictions for the relationship between external details and gestures, we sought a relationship with the non-representational (beat) gestures because they are discussed in the context of discourse processes. Confirming this, for the past events, non-representational gestures were correlated with external details ($\tau = .35$, $p = .002$); specifically with personal semantic details, namely AF ($\tau = .27$, $p = .02$) and SK ($\tau = .50$, $p < .001$).

For the future autobiographical events, as presented in Table 3 we did not observe an association between representational gesture use and episodic (internal) details ($\tau = .10$, $p = .39$). However, the analysis revealed that both representational ($\tau = .26$, $p = .02$) and non-representational gestures were related to external details ($\tau = .27$, $p = .02$). Non-representational gestures were specifically correlated with repetition ($\tau = .29$, $p = .02$) and SK ($\tau = .35$, $p = .005$). Overall, in both past and future events, external details correlated with non-representational

**Table 2. Correlation matrix for past events.**

| | | Representational | Non-representational | Total Gesture | Gesture per Utterance |
|---|---|---|---|---|---|
| Representational | τ | — | | | |
| | p | — | | | |
| Non-representational | τ | .299 | — | | |
| | p | .007 | — | | |
| Total Gesture | τ | .593 | .715 | — | |
| | p | < .001 | < .001 | — | |
| Gesture per Utterance | τ | .514 | .540 | .655 | — |
| | p | < .001 | < .001 | < .001 | — |
| Internal Total | τ | .271 | -.033 | .131 | .128 |
| | p | .016 | .768 | .240 | .249 |
| Internal-Place | τ | .275 | .086 | .180 | .196 |
| | p | .024 | .482 | .134 | .103 |
| Internal-Perceptual | τ | .361 | .071 | .237 | .225 |
| | p | .002 | .551 | .044 | .056 |
| External Total | τ | .149 | .349 | .331 | .065 |
| | p | .180 | .002 | .003 | .551 |
| External-AF | τ | .165 | .271 | .277 | .132 |
| | p | .153 | .020 | .016 | .249 |
| External-SK | τ | .066 | .504 | .366 | .136 |
| | p | .587 | < .001 | .002 | .257 |
| MTT | τ | .180 | .251 | .268 | .236 |
| | p | .143 | .043 | .028 | .053 |
| Reliving | τ | .292 | .244 | .325 | .246 |
| | p | .016 | .045 | .007 | .040 |
| Vividness | τ | -.057 | .124 | .112 | .076 |
| | p | .646 | .320 | .366 | .536 |
| CBT | τ | .084 | .103 | .119 | .056 |
| | p | .471 | .378 | .304 | .623 |
| MRT | τ | .088 | -.020 | .041 | .105 |
| | p | .430 | .857 | .710 | .344 |

gestures (i.e., beat). Lastly, a Fisher's Z-transformation revealed that the association between representational gestures and internal details for past and future conditions are not significantly different ($z = 1.20$, $p = .23$).

An additional dimension in exploring the relationship between episodicity and gesture use in event narrations is the subjective sense of recollection. In other words, are representational gestures related to the phenomenology of autobiographical memories? First, we sought to investigate the association between overall gesture rate and subjective phenomenology. For the past events, we observed that gesture per utterance only correlated with reliving ($τ = .25$, $p = .04$), whereas the association between vividness ($τ = .08$, $p = .54$) and mental time travel ($τ = .24$, $p = .053$) did not reach significance. Regarding the phenomenological ratings, we observed that the feeling of reliving was correlated with both representational ($τ = .29$, $p = .02$) and non-representational gestures ($τ = .24$, $p < .05$) in past events. Also, the feeling of mental time travel was correlated with non-representational gestures ($τ = .25$, $p = .04$), suggesting that increased use of gestures is benefiting the reliving component of the autobiographical memories. In contrast to past events, neither the gesture categories nor gesture per utterance correlated with the

**Table 3. Correlation matrix for future events.**

| | | Representational | Non-representational | Total Gesture | Gesture per Utterance |
|---|---|---|---|---|---|
| Representational | τ | — | | | |
| | p | — | | | |
| Non-representational | τ | .342 | — | | |
| | p | .002 | — | | |
| Total Gesture | τ | .671 | .695 | — | |
| | p | < .001 | < .001 | — | |
| Gesture per Utterance | τ | .502 | .608 | .662 | — |
| | p | < .001 | < .001 | < .001 | — |
| Internal Total | τ | .098 | .039 | .079 | -.039 |
| | p | .389 | .734 | .483 | .726 |
| External Total | τ | .257 | .274 | .289 | .074 |
| | p | .023 | .015 | .010 | .506 |
| External-SK | τ | .208 | .350 | .296 | .130 |
| | p | .094 | .005 | .016 | .287 |
| External-Repetition | τ | .223 | .289 | .279 | .189 |
| | p | .078 | .023 | .025 | .127 |
| MTT | τ | -.070 | -.092 | -.071 | -.198 |
| | p | .568 | .454 | .561 | .099 |
| Reliving | τ | .102 | .023 | .073 | .075 |
| | p | .388 | .845 | .535 | .520 |
| Vividness | τ | -.053 | .103 | .035 | .027 |
| | p | .663 | .397 | .769 | .824 |
| MRT | τ | .097 | .033 | .063 | .059 |
| | p | .391 | .769 | .573 | .596 |
| CBT | τ | -.003 | .056 | .007 | .000 |
| | p | .982 | .630 | .954 | 1.000 |

phenomenology ratings in future event narrations (for the gesture per utterance: reliving, $p$ = .52; vividness, $p$ = .82; mental time travel, $p$ = .09, as presented in Table 3).

## Discussion

In the present study, we investigated the relationship between co-speech hand gestures -particularly representational gestures- and episodic autobiographical event representations. We asked participants to narrate past autobiographical, future autobiographical, and non-autobiographical events that differ in terms of their level of episodicity. Our study yielded a number of findings: First, we did not observe gesture rate differences across event conditions. Second, and most central to our current goal, we found that representational gesture production was associated with the specific episodic details as measured by the Autobiographical Interview [37]. Third, we have also observed a correlation between gesture use and subjective reports of recollective phenomenology in the past event condition. Surprisingly, this association holds both for representational and nonrepresentational gestures. Finally, non-representational gesture use and non-episodic information in the narratives of past and future episodic events were associated.

Regarding the first finding, our reasoning was that preparing to narrate an autobiographical event (as opposed to a non-autobiographical one) would trigger gesture use (due to the specific events' multimodal representation) and gesture use would, then, represent and activate

additional episodic information. Surprisingly, we did not observe differences in the rate of gesturing across these conditions. One possible explanation may lie in our choice of method in recruiting the events. Since we used spatial cue words (e.g., street) to sample the autobiographical events and spatial contexts (e.g., bank transaction) for the non-autobiographical conditions, the participants may have treated the events to be narrated as more scene-based than event-based [58]. when constructing them. For instance, this might have resulted in future events to resemble non-autobiographical scenes rather than autobiographical/episodic events (see Table 1 for gesture rates in conditions). Furthermore, given the evidence on the recruitment of gestures during spatial processing (e.g., [59–61]), the added scene-based treatment may have rendered the three conditions more similar in terms of the need to engage representational gestures. Given that in a previous study, participants gestured more in telling frightening experiences than when they were telling how to do grocery shopping [36], future designs need to vary the nature of autobiographical events to better outline the contribution of gestures.

Despite the lack of differences in the rate of gesture use across conditions, one finding lends strong support for an association between gesture use and episodic details: In the past event condition, we found a positive correlation between the episodic (internal) details and representational gestures. While the mechanism underlying these correlations between internal/episodic details and representational iconic gesture use is not completely clear based on the current design, our finding that representational gesture use was positively correlated with the existence of sensory details and place-related details suggests that gestures may indeed help with maintaining spatial and sensory information associated with the event representation (e.g., [9]). This is an important finding for the event construction literature according to which an image of the location serves as a scaffold for elaboration with additional details [30, 62]. If gestures support representing the spatial context in autobiographical episodic memory, it is possible that gesturing would have a corresponding benefit in recalling additional episodic details. This is the first study to our knowledge that sought a relationship between episodic informational units (e.g., sensory details) and gesture production. An alternative proposal to characterize the relationship between gestures and episodic representations reverses the directionality by arguing that gestures are supported by hippocampal episodic event representations [36]. Support for this argument comes from the finding that gesture use was compromised in patients with hippocampal damage, such that across different narratives (i.e., procedural and episodic) patients gestured less compared to the healthy group. The level of episodic details and gesture frequency was associated in a small group of neurotypical participants ($N = 9$); however, no such link was observed in the patient group. Although it is intuitive to think that gestures emerge from episodic representations, the evidence is not conclusive based on Hilverman et al.'s [36] selection of methods for two reasons: (1) Their choice of autobiographical events was emotional in nature; namely, the most frightening experience and personal context of JFK's assassination. Given that hippocampus is also implicated in emotional processing [24, 63, 64], it is not possible to tease out whether it is the episodicity or the emotionality of the events driving the effects. (2) Their control -procedural event- condition included imagining scripted events that were non-episodic in nature. However, it is not possible to disentangle whether less frequent gesturing is due to the lack of episodic quality of the events or the fact that the participants needed to construct these events rather than remembering them. A good control condition would be to also include an imagined autobiographical event condition where construction processes would still be at work, but the content would be episodic. All in all, Hilverman's finding that gestures are sensitive to episodic memory representations presents an interesting case for our purposes. Others have also alluded that the relationship between gesture use and cognitions is bidirectional [33]. We contend that "episodicity" of the

narrative may trigger gesture production, and gestures in turn may guide the retrieval of further episodic details.

Interestingly, however, the association between gesture use and episodic details was observed only for the past autobiographical events. Two factors might have contributed to the lack of association between internal details and gesture use in future events. The narratives participants produced were rather short to allow for enough episodic elements, either in gesture or verbal form. Second, as Hostetter and Alibali [35] showed when verbally describing images, individuals gestured more than when they had previous physical experiences with the stimuli in the images. Therefore, since our participants did not have previous sensory and motor experience with the information they were describing in the future autobiographical event condition, gestures could not be effective in helping speakers sustain episodic information stored in memory to the extent that they did in the past event condition. Although gesture use can activate, manipulate, package, and explore spatio-motoric explanations and in turn help schematize information [2], it can be argued that particularly activation and manipulation of sensory and motor information is stronger for the past event representations than future event representations. Gesture use may activate the already established spatio-motoric event details better than formulating new event details. Then, when asked to imagine a personal event, not having a past sensory experience may lead to a less strong connection to the episodic details.

Furthermore, Addis [18] rightfully hinted that, in addition to the usual episodic elements, future event representations may also include memory for the imagination process itself. For example, engaging in reasoning or changing content to be more plausible would be manifested in external details, maybe in the form of repetitions or general semantics. This may have contributed to the unexpected relation between representational gestures and external events such that when information is captured and packaged in representational gestures, the freed-up cognitive resources may be dedicated to the discourse processes, i.e., the use of external details, in order to select and contextualize the internal event.

With regards to phenomenology, we found that gesture use was related to the sense of reliving the past events however this was not specific to the representational gestures as we predicted. Non-representational gesture use was also correlated with the subjective sense of recollection. Although prior work has not tested this relationship, the evidence on gestures alleviating cognitive load in narrative recall and problem-solving (e.g., [8, 65–67]) suggests that reduced cognitive demands during retrieval via the use of gestures may lead to richer reflective recollective experience.

Even though we did not have specific predictions regarding non-representational gestures, we did find rather strong correlations between nonrepresentational gestures and the external details in both past and future events. A tentative explanation relies on the possible communicative role played by both the non-representational gestures and external details. The beat (non-representational) gestures are thought to capitalize on communicative functions, such as fluency in discourse, rather than representative functions [68, 69]. Interestingly, evidence from the autobiographical literature also suggests that the reason external details, and therefore, semantic information, are included in the autobiographical narratives is to elaborate or "embellish" the discourse [70 (p. 1)]. It is not surprising, then, when non-representational gestures assume a fluency-resolving role, they would enhance the generation of semantic information in autobiographical narratives. Further studies are needed to establish whether there is a mapping between the semantic information in autobiographical memories and beat gestures or whether beat gestures are more frequently used during the search for specific episodic information.

Given this backdrop, the present findings provide good evidence that gestures support episodic event representations. Their role seems to be stronger in lived past experiences. To

further qualify the relationship, future research is needed by carefully manipulating the gesture use (i.e., instructed gesture use) as well as the sampling of episodic events. The present study was a first step in establishing the contribution of co-speech gestures to autobiographical memories as complex mental representations. Future studies can be designed to further test the specific predictions based on the gesture-for-conceptualization framework [2]. In particular, the role of gestures' specific functions (activation, manipulation, packaging, and exploring spatio-motoric information) for past and future event representations can be investigated. Future studies should also replicate the findings in other languages. Even though prior work has successfully used the Autobiographical Interview in Turkish (e.g., [31]), the way gestures represent concepts such as time and space, may be different across languages (e.g., [71, 72]) which may have implications on their influence on memory retrieval. Last, although our findings did not provide support for the role of visuo-spatial working memory and spatial skills on gesture use in narrating autobiographical memories, future research should also examine the role of individual differences in using cognitive resources, both for visuo-spatial and verbal skills, on episodicity in more detail [73].

The present research also represents a fertile research area with practical implications. A growing literature has shown that reduced episodic specificity (memories with a low level of episodic details) is associated with psychological health (e.g., [74]) as well as Alzheimer's Disease (e.g., [75]). Since most of these studies are concerned with verbal and written accounts of episodic events, investigating gestures' role in contributing to episodic specificity of memories would be critical. Given the evidence that information in gesture is sometimes repeated in speech, and sometimes only reserved for gesture (e.g., [76]), future work examining whether episodic details in gestures and in speech complement and influence each other would be a potential addition to the existing efforts in designing intervention programs to induce episodic specificity [77].

In conclusion, this study presents one of the first pieces of evidence about the role of representational gestures on episodic event details, such as sensory and place-related details and recollective quality in autobiographical memories. These associations were specific to recalling past events rather than imagining future events or non-autobiographical events. Our findings highlight the potential of gesture production in maintaining spatial and sensory information associated with the event representation that in turn aids the retrieval of details in autobiographical memories.

## Acknowledgments

We would like to thank research assistants Nesli Aslan, Deniz Alp Domaniç, Berkay Toksöz, Lal Atatüre, Cansu Ağaoğlu, and Belgin Deryalar for their assistance in participant recruitment, data collection, and coding gestures. We would also like to thank the reviewers for their valuable feedback on this paper.

## Author Contributions

**Conceptualization:** Cagla Aydin, Tilbe Göksun.

**Data curation:** Cagla Aydin, Tilbe Göksun, Ege Otenen, Selma Berfin Tanis.

**Formal analysis:** Cagla Aydin, Tilbe Göksun, Ege Otenen, Selma Berfin Tanis, Yağmur Damla Şentürk.

**Investigation:** Cagla Aydin, Tilbe Göksun, Ege Otenen, Selma Berfin Tanis, Yağmur Damla Şentürk.

**Methodology:** Cagla Aydin, Tilbe Göksun, Ege Otenen, Selma Berfin Tanis, Yağmur Damla Şentürk.

**Project administration:** Cagla Aydin, Tilbe Göksun.

**Resources:** Cagla Aydin, Tilbe Göksun.

**Supervision:** Cagla Aydin, Tilbe Göksun.

**Validation:** Cagla Aydin, Tilbe Göksun.

**Visualization:** Cagla Aydin, Tilbe Göksun.

**Writing – original draft:** Cagla Aydin, Tilbe Göksun, Ege Otenen, Selma Berfin Tanis, Yağmur Damla Şentürk.

**Writing – review & editing:** Cagla Aydin, Tilbe Göksun, Ege Otenen, Selma Berfin Tanis, Yağmur Damla Şentürk.

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
