## [Decision Letter · Decision Letter 0]

10 Oct 2022

PONE-D-22-18508The role of gestures in autobiographical memory

PLOS ONE

Dear Dr. Aydin,

Thank you for submitting your manuscript to PLOS ONE. After careful consideration, we feel that it has merit but does not fully meet PLOS ONE’s publication criteria as it currently stands. Therefore, we invite you to submit a revised version of the manuscript that addresses the points raised during the review process.

Three  reviewers and myself have read  your manuscript. We are all in agreement that it describes novel research concerning the link between embodiment and autobiographical memory. However a number of points were raised that would enhance the quality of the paper and are therefore required in a revision. I highlight  some of the key points raised but you need to address all points in your revision.   The introduction needs to revised to  present greater evidence for the role of embodiment in remembering. A key point in the  introduction  is that the visuospatial content of speech  predicts gesture use.The coherence of the introduction would improve if this point was made earlier. Further .consider if your hypotheses should be directional given the novelty of the work. Several reviewers requested justification for the sample size. Issues were also raised about your method of calculating inter rater reliability.  You should use cohens k or kappa. Further the results were hard to follow. Have a  data analysis section where you outline the statistical tests that will be used to test your hypotheses. . Also consider using a linear mixed model to predict detail (or rating) from gesture use as a function of condition, gesture type, and detail type rather than running separate correlations. For the discussion it may also be relevant to discuss if the gesture-conceptualization framework may or may not account for the relationship between representational gestures and external details in terms of schematized self-knowledge which is observed in both past and future thinking in the present study (Kita et al. 2017) Please submit your revised manuscript by November15 2022.If you will need more time than this to complete your revisions, please reply to this message or contact the journal office at plosone@plos.org. Please include the following items when submitting your revised manuscript:

A rebuttal letter that responds to each point raised by the academic editor and reviewer(s). You should upload this letter as a separate file labeled 'Response to Reviewers'.A marked-up copy of your manuscript that highlights changes made to the original version. You should upload this as a separate file labeled 'Revised Manuscript with Track Changes'.An unmarked version of your revised paper without tracked changes. You should upload this as a separate file labeled 'Manuscript'

We look forward to receiving your revised manuscript.r

Kind regards,

Barbara Dritschel, PhD

Academic Editor

PLOS ONE

Journal Requirements:

4. We note that you have stated that you will provide repository information for your data at acceptance. Should your manuscript be accepted for publication, we will hold it until you provide the relevant accession numbers or DOIs necessary to access your data. If you wish to make changes to your Data Availability statement, please describe these changes in your cover letter and we will update your Data Availability statement to reflect the information you provide

Reviewers' comments:

Reviewer's Responses to Questions

**Comments to the Author**

1. Is the manuscript technically sound, and do the data support the conclusions?

Reviewer #1: Yes

Reviewer #2: Partly

Reviewer #3: Yes

2. Has the statistical analysis been performed appropriately and rigorously? 

Reviewer #1: Yes

Reviewer #2: I Don't Know

Reviewer #3: Yes

3. Have the authors made all data underlying the findings in their manuscript fully available?

Reviewer #1: Yes

Reviewer #2: Yes

Reviewer #3: Yes

4. Is the manuscript presented in an intelligible fashion and written in standard English?

Reviewer #1: Yes

Reviewer #2: Yes

Reviewer #3: Yes

5. Review Comments to the Author

Reviewer #1: The role of gestures in autobiographical memory

This manuscript reports on a behavioural study that tested the associations between gesture use (and types of gestures) and the details used for and experience of constructing autobiographical and non-autobiographical mental representations. The main finding was that gesture use was consistently engaged across three forms of mental representations (past, future, non-autobiographical) but related only to episodic details and experiential ratings for the past autobiographical events. Reasons for these findings are discussed at length.

I enjoyed reading this paper. I thought this it was a very well-written manuscript that tackled an interesting subject in the field of autobiographical memory research. The rationale is clear, the methods are well-described, and the discussion was thorough. I liked that the authors explored alternate explanations for their results.

I do have a few suggestions to improve the paper that are listed by section below:

Introduction. It would be great if there was a bit more information for the novice reader. First, it would be nice to expand slightly on the evidence of embodied forms of remembering in the introduction. Second, since the paper is following the categorization of iconic and metaphoric gestures without deictic as representational and beat gestures as non-representational, could they elaborate on how this categorization was made?

The hypothesis is that certain types of gestures lead to more episodically-detailed representations, but the authors don’t test directionality with their design. For this reason, they state an argument against the alternative proposal that gestures are supported by hippocampal episodic event representations in the introduction. However, I wonder if they can simply say that their study was to first test the association between gestures and episodic representations across various forms of representations in order to gain support for their hypothesis in in the introduction (as they do on page 8, but perhaps state this earlier). Then, they review the alternate interpretations in the discussion. I suggest this because, even if this alternate proposal is rejected, the study is still correlational in nature, meaning one can’t say what is causing what.

Methods: Can the authors justify their sample size? Were there any sex and gender effects?

It would be great to have more information on the gesture coding, perhaps with images. Were these scored by multiple raters and if so, what was the inter-rater reliability?

Analysis: When the details and subjective phenomenology ratings were correlated with gesture per utterance, were there corrections for multiple comparisons? As well, the authors should test whether there is a difference between reported significant correlations (e.g., between detail and gesture use in past autobiographical events) and non-significant correlations (e.g., between detail and gesture use in future autobiographical events) with statistics (e.g., Fisher’s transformation). This is because such differences in significance are interpreted in the discussion. I do wonder if a better statistical approach would be to use a linear mixed model to predict detail (or rating) from gesture use as a function of condition, gesture type, and detail type rather than running separate correlations? This would allow the authors to make more solid conclusion about differences in how these variables estimate detail use.

Discussion: One issue to discuss is finding that past, future autobio and non-autobio events related to gesture use and then finding/discussing how details correlate the gesture use only for the autobiographical condition. The pairing of these findings suggest to me that the correlations reported could reflect a measure of on-topic content (ie., internal details for autobio events) relating to gesture use. Would there be a way to test this or perhaps speak to this in the discussion? Perhaps relation to broader embodied cognition work in the discussion as well.

Reviewer #2: The manuscript describes a study investigating the relationship between gesture and autobiographical event (re)construction. By integrating standard research methods for assessing and coding for autobiographical event processing (narration of past, future and non-autobiographical events) and gesture (representational and non-representational) the authors have developed a novel and creative paradigm for further investigating features of embodied cognition. As predicted, the authors identified a relationship between the frequency of representational gestures and episodic features of autobiographical events. Namely, that representational gesture was associated with the frequency of internal details and reliving in past autobiographical events. In addition, relationships between representational and non-representational gestures and external details were observed for both past and future autobiographical events. Overall the topic area is novel and highlights potential new avenues of research into embodied cognition. The introduction provides a solid theoretical basis for conducting the research and clear potential hypotheses to be investigated. However, a number of points should be addressed to improve the clarity and understanding of this study prior to publication. I have outlined these points in more detail below.

Calculation of participant sample size and presentation of associated analyses

The basis for the sample size selected for this study should be stated, it is currently unclear if the sample size is based on any a priori power analyses or previous literature in the field. This is of particular relevance in relation to the null findings when examining the first hypothesis comparing overall gesture rate between the three autobiographical conditions. It is relevant to determine if null findings could be a result of low power. If this is a possibility, it may be relevant to state this in the discussion. Relatedly, the specific statistical analyses conducted to test this hypothesis, and the role of visuospatial skills and spatial imagery skills are not reported. There are also minor errors in the Note in Figure 1 (“None-ABM) and the graph legend replicates information already presented in the figure.

Coding of event and gesture narratives: Interrater reliability

Percentage agreement between raters was used as a measure of interrater reliability for the event narratives. This measure of reliability has been known to overestimate the level of agreement between raters. Inter-rater reliability statistics such as Cohens Kappa may provide a better estimate of interrater reliability. It is also unclear if interrater reliability was conducted for the gesture coding?

The structure and organization of the results

While the results are interesting, this section of the manuscript is very difficult to follow, as information is missing, reported several times and lacks clarity in places. It may be worthwhile to include a statistical analyses section at the start of the results to present the rationale for the selection of analyses conducted to the test the hypotheses outlined and any additional exploratory analyses. Presenting the correlations between event details and gesture in the form of a table may also improve the clarity of the results, and ensure that all relevant correlations are included.

As outlined above, the type of statistical analyses conducted to test the first hypothesis are not stated explicitly. The statistics for a number of relevant correlations are also missing. For example, the correlation between representational gesture and external details and the non-significant correlations between gesture rate in future events and phenomenology (which are stated but the statistics are not reported). In contrast, other correlations, such as the correlation between representational details and external details in future events are reported twice. It is also unclear as to why the correlations between total number of gestures and event details are included and what additional value they provide.

Discussion

In the discussion the findings related to the relationship between representational gestures and episodic detail are clearly presented and the associated links to previous research and future implications described. However, it may also be relevant to discuss if the gesture-conceptualization framework may or may not account for the relationship between representational gestures and external details in terms of schematized self-knowledge which is observed in both past and future thinking in the present study (Kita et al. 2017).

Minor points

P11, line 219 the term “atemporal” is used, it is not clear what this means in the context of the non-autobiographical event.

P11 lines 225-226 suggest that only the scales for vividness, reliving and mental time travel were utilized in the present study. However, in the results p 15, line 30, statistics related to emotional valence and intensity are reported. Were the measures reported in this manuscript (reliving, vividness and mental time travel) a subset of a larger number of items included the study which also included valence and intensity?

These results on valence and intensity are not reported elsewhere in the paper so I was unsure of their relevance. It was interesting to note that valence negatively correlated with total number of gestures per utterance.

P7, line 128, the authors use the term “episodicness” and I wonder if the term episodicity may be more appropriate (see Habermas & Diel, 2013)

P19, line 374-376 the authors suggest that future events may resemble non-autobiographical events to a greater extent than anticipated. One way to examine the content of the events reported would be to code and compare the frequency of internal and external details in the events. A lower level of internal details in the non-autobiographical events may support the validity of the differentiating the autobiographical and non-autobiographical events in terms of autobiographical content.

Habermas, T., & Diel, V. (2013). The episodicity of verbal reports of personally significant autobiographical memories: Vividness correlates with narrative text quality more than with detailedness or memory specificity. Frontiers in Behavioral Neuroscience, 7, 110.

Reviewer #3: This is the first study I know of that looks at gesture use in autobiographical memory in adults. So many previous studies on gestures have focused on cartoon retelling tasks, this is a breath of fresh air! It was an interesting approach to look at the episodic specificity as a predictor of gesture production. The null findings for the cognitive abilities predicting individual differences also adds to the literature.

The framing of the article could be tightened on two grounds: 1) how the genre might make a difference and 2) how different gesture types might matter. The introduction does get to the point that it might be the visuospatial content of speech that predicts gesture use. The introduction could get there faster. As for gesture type, it is not entirely clear why non-representational gestures were included in the analyses (yes, the results were intriguing, but it is not clear how to interpret them!).

Given the wide individual variability in gesture rate reflect an individual tendency to gesture a lot (or a little)? In other words, were there correlations in gesture rate across conditions? If so, that would support the argument that gesturing reflects (at least in part) an individual’s characteristics. Even if no support for that individual aspect being visuospatial ability was found in this study.

Somewhat smaller points:

-There is no exposition of the linguistic construction that was used as the baseline (either words or utterances). I think it was words, but were false starts and self-repetitions counted? Why or why not? And words were orthographic words? If utterances, then what was the definition of “utterance”?

- I don’t understand why participants who did not gesture at all were not included in the analyses. At least one study (with children) found that not gesturing led to reduced visuospatial content (Laurent et al., 2020).

-It could make the results easier to follow to include a table with all the correlations.

-Were any corrections be made for multiple correlations?

-How was the subjective sense of recollection measured?

-I don’t know if it helps at all, but I know of at least one study that included autobiographical memories with children (Marentette et al., 2020). It might not be useful since the focus of that study was different. That said, if I remember correctly, the gesture rate was lower when the children told autobiographical stories than when they told fictional stories.

Very small points:

p. 3, second paragraph, line 2: what makes a ‘type’ of gesture?

p. 5, starting line 4: It was unclear whether the participants in Cook et al. (2010) gestured or if the events themselves were highly visuospatially imagistic

p. 5, sentence starting on line 4 from the bottom: later on in the paper, the authors make the argument that the directionality might be bidirectional.

p. 12, last sentence: how were these percentages calculated? Particularly, what was in the denominator?

p. 13, first line of text: what does “representative” mean? Were self-adaptors included if they were rhythmic?

p. 13: note that the description here is in terms of gestures per words while the Figure shows gestures per utterance.

p. 13, last line: explain what the post-hoc analysis was exactly.

p. 16, midway through the first paragraph bout the correlation between representational gestures and internal details. Was this correlation with the NUMBER of gestures or the gesture RATE? If the former, couldn’t this correlation come about because people who talked more gestured more?

p. 16, for the correlations that the authors would really like to remain salient in readers’ minds, scatterplots could be useful.

p. 19, first paragraph: it would be useful if the authors reported the number of details across conditions in the results section so readers can follow their arguments here.

p. 22, last sentence: this point was already made

References

Laurent, A., Smithson, L., & Nicoladis, E. (2020). Gesturers tell a story creatively; non-gesturers tell it like it happened. Language Learning and Development, 16(3), 292-308.

Marentette, P., Furman, R., Suvanto, M. E., & Nicoladis, E. (2020). Pantomime (not silent gesture) in multimodal communication: evidence from children’s narratives. Frontiers in psychology, 11, 575952.

6. PLOS authors have the option to publish the peer review history of their article (what does this mean?). If published, this will include your full peer review and any attached files.

Reviewer #1: No

Reviewer #2: No

Reviewer #3: **Yes: **Elena Nicoladis

---

## [Author Response · Author response to Decision Letter 0]

19 Nov 2022

Dear Editor,

We thank you for providing us with this opportunity to improve on the quality of our manuscript based on the rich set of comments provided by the Editor and the Reviewers. Below, we provide detailed point-by-point responses (italicized) to these comments. The references that we cite are listed at the end of this letter.

Along with our individual responses to each of the reviewer comments below, we would like to summarize the main changes in the revised version of the manuscript for your attention.

“Three reviewers and myself have read your manuscript. We are all in agreement that it describes novel research concerning the link between embodiment and autobiographical memory. However a number of points were raised that would enhance the quality of the paper and are therefore required in a revision. I highlight some of the key points raised but you need to address all points in your revision. The introduction needs to revised to present greater evidence for the role of embodiment in remembering. A key point in the introduction is that the visuospatial content of speech predicts gesture use.The coherence of the introduction would improve if this point was made earlier. Further .consider if your hypotheses should be directional given the novelty of the work. Several reviewers requested justification for the sample size. Issues were also raised about your method of calculating inter rater reliability. You should use cohens k or kappa. Further the results were hard to follow. Have a data analysis section where you outline the statistical tests that will be used to test your hypotheses. Also consider using a linear mixed model to predict detail (or rating) from gesture use as a function of condition, gesture type, and detail type rather than running separate correlations. For the discussion it may also be relevant to discuss if the gesture-conceptualization framework may or may not account for the relationship between representational gestures and external details in terms of schematized self-knowledge which is observed in both past and future thinking in the present study (Kita et al. 2017).”

Response:

Thank you very much for your overall encouraging evaluation.

1. We revised the Introduction, especially the beginning paragraphs, to reflect greater evidence for embodied remembering. A main argument being the association between the visuospatial nature of the representational gestures and the visuospatial content of episodic autobiographical memories appears earlier in the main text now (p.4, end of the second paragraph of the Introduction). After stating that earlier, our hypotheses are more directional (i.e., we moved the alternative explanations for the association to the Discussion -given the idea is novel, as you stated.)

2. We added justification for the sample size. 

3. For the gesture coding interrater reliability, we added the kappa values. For the speech codings, since there were multiple code categories, following the standard procedure in the Autobiographical Interview coding, we reported intraclass correlation coefficients.

4. Alerted by your and Reviewer 2’s concerns, the Results section now has an Analytical Approach section where we list and describe the statistical tests conducted. The order of presentation of the findings is changed accordingly. We first discuss gesture rate differences across conditions. The examination of individual difference variables follows. Next, we report the associations between internal details and gestures for past and future events. And finally, we report the association of gestures with the phenomenological properties of the memories. Also, we now added two tables which summarize all the correlations as per reviewer request. We also discussed whether to use a linear mixed model. Given the fact that our predictions were rather exploratory (an association is being established for the first time) than confirmatory, we concluded that independent correlations may be more suitable to the readers. If the editor and the reviewers think that is needed and helpful, we may switch to that approach. We also want to emphasize that we statistically compared the correlations of representational gesture use in past and future conditions with Fisher’s transformation in line with the Reviewer 2’s suggestions.

5. Our findings did not reveal an association between representational gestures and external details in the past condition. We, however, observed such an association in the future condition. Gesture-for-conceptualization framework proposes that gestures help schematizing information by way of assuming different functions, such as activating, manipulating, packaging and exploring information. Therefore, it is possible that a higher number of non-episodic elements, such as external details, to be present in the narratives with increased gesture use because external details are thought to reflect schematic information. However, since this was a posthoc elaboration, and was not confirmed in the past condition, we resort to add it as future direction 

Reviewer #1: 

This manuscript reports on a behavioural study that tested the associations between gesture use (and types of gestures) and the details used for and experience of constructing autobiographical and non-autobiographical mental representations. The main finding was that gesture use was consistently engaged across three forms of mental representations (past, future, non-autobiographical) but related only to episodic details and experiential ratings for the past autobiographical events. Reasons for these findings are discussed at length.

I enjoyed reading this paper. I thought this it was a very well-written manuscript that tackled an interesting subject in the field of autobiographical memory research. The rationale is clear, the methods are well-described, and the discussion was thorough. I liked that the authors explored alternate explanations for their results.I do have a few suggestions to improve the paper that are listed by section below:

Thank you very much for your overall encouraging evaluation.

COMMENT 1:

Introduction. It would be great if there was a bit more information for the novice reader. First, it would be nice to expand slightly on the evidence of embodied forms of remembering in the introduction. Second, since the paper is following the categorization of iconic and metaphoric gestures without deictic as representational and beat gestures as non-representational, could they elaborate on how this categorization was made?

RESPONSE:

We thank the reviewer for pointing out two important issues. Regarding the first one, we revised the very first paragraph of the Introduction (p. 3), and talked about some evidence on embodied remembering, and how they complement the present idea (gesture & memory relationship) well. 

The paragraph now reads as:

“Abundant research has suggested that gestures are not just used in the service of communication (e.g., Hostetter & Alibali, 2019) but are also involved in the speakers’ cognitive processes and mental representations (e.g., Kita et al., 2017). Autobiographical memories are complex mental representations of the self that are constructed from different types of episodic and semantic knowledge, and thought to contain sensorimotor elements (Conway, 2001). Provided that autobiographical memories are not fully amodal representations, here, we seek a link between gesture production and autobiographical memory representations. This idea complements well with the embodied cognition literature, where memory retrieval has been shown to benefit from sensorimotor activations during encoding (e.g., Dijkstra & Zwaan, 2014). It has been shown that embodied forms of remembering involve different modalities, including eye-movements (e.g., Laeng & Teodorescu; 2002), body postures (e.g., Dijkstra et al., 2007); and movements (e.g., Casasanto & Dijkstra, 2010). Similarly, the beneficial role of gesture production has been shown during episodic memory (e.g., Cook et al., 2010; Wesp et al., 2001) and autobiographical memory retrieval (Marentette et al., 2020). However, to our knowledge, none of the studies attempted to explore the possible contribution of gesture production during the retrieval of complex autobiographical representations in adults. Given this backdrop (e.g., Dijsktra et al., 2007; Iani, 2019, for a review), we aim to examine the link between gesture production and autobiographical memory representations.”

With respect to the second point, we also agree that it would be better to give more detail about gesture categorization in the Introduction. Therefore, we now elaborate on the reasoning behind this categorization and cited the relevant research (e.g., Arslan & Göksun, 2021, 2022) in the corresponding paragraph (p. 4). 

“Co-speech gestures are intertwined with speech, temporally and semantically (McNeill, 1992). Gestures are mainly considered in four categories: iconic gestures depict action, motion, and shape or indicate location and trajectory; and metaphoric gestures represent abstract ideas. Beat gestures, on the other hand, are simple motoric movements produced along with the rhythm of the speech. Last, deictic gestures refer to pointings of objects or abstract entities (McNeill, 1992). Recent frameworks mainly consider iconic and metaphoric gestures without deictic ones as representational and beat gestures as non-representational (Hostetter & Alibali, 2019; Kita et al., 2017). In this paper, we follow the same categorization as we focus on specific representations of concrete and abstract concepts in iconic and metaphoric gestures (see Arslan & Göksun, 2021, 2022 for similar categorizations). Since representational gestures are visuospatial in nature (Kita et al., 2017), this categorization will allow us to investigate the possible contribution of visuospatial information to autobiographical remembering.”

COMMENT 2: 

The hypothesis is that certain types of gestures lead to more episodically-detailed representations, but the authors don’t test directionality with their design. For this reason, they state an argument against the alternative proposal that gestures are supported by hippocampal episodic event representations in the introduction. However, I wonder if they can simply say that their study was to first test the association between gestures and episodic representations across various forms of representations in order to gain support for their hypothesis in the introduction (as they do on page 8, but perhaps state this earlier). Then, they review the alternate interpretations in the discussion. I suggest this because, even if this alternate proposal is rejected, the study is still correlational in nature, meaning one can’t say what is causing what.

RESPONSE: 

Yes, we also thought this change would tighten the text. Therefore, in the revised version, it is stated much earlier (p.5., end of the first paragraph) that the present study was the first study to target this association. Then, we discussed the alternative proposals (e.g., Hilverman et al.) in the Discussion section.

COMMENT 3:

Methods: Can the authors justify their sample size? Were there any sex and gender effects?

It would be great to have more information on the gesture coding, perhaps with images. Were these scored by multiple raters and if so, what was the inter-rater reliability? 

RESPONSE:

We apologize for not being clear on how we decided on the sample size. Our sample size is determined by following the previous studies in the literature (e.g., Cook et al. 2010; Cook & Tanenhaus; 2009; Church et al., 2007; Ping et al., 2014). Additionally, post-hoc power analysis suggests that our sample of 41 people successfully detects the power of .85, with a false positive rate of .05 under the estimated effect size of the study (d = 0.44). This information was now added under Method → Participants on page 10.

Another point raised by the reviewer is regarding the age and gender differences in the findings. We agree that observing such differences would be intriguing, however we did not initially hypothesize to see gender and age differences. Additionally, our current sample is not convenient to make such claims since it is not a diverse sample but dominated by undergraduate-level female students (Mage = 19.5, SDage = 0.7; 60% female). 

Lastly, we revised the section where we explained gesture coding (p. 13). As suggested by the reviewer, we now added gesture examples and inter-rater reliability information in the corresponding paragraph. 

COMMENT 4:

Analysis: When the details and subjective phenomenology ratings were correlated with gesture per utterance, were there corrections for multiple comparisons? As well, the authors should test whether there is a difference between reported significant correlations (e.g., between detail and gesture use in past autobiographical events) and non-significant correlations (e.g., between detail and gesture use in future autobiographical events) with statistics (e.g., Fisher’s transformation). This is because such differences in significance are interpreted in the discussion. I do wonder if a better statistical approach would be to use a linear mixed model to predict detail (or rating) from gesture use as a function of condition, gesture type, and detail type rather than running separate correlations? This would allow the authors to make more solid conclusions about differences in how these variables estimate detail use.

RESPONSE:

As noted in our response to the Editor, we considered whether to switch to a linear mixed model. Given that our predictions were rather exploratory (an association is being established for the first time) than confirmatory, we concluded that independent correlations may be more suitable to the readers from different backgrounds (gesture or memory researchers). If the editor and the reviewers think that is needed and helpful, we may switch to that approach. 

As per the reviewer’s suggestion, we employed Fisher’s transformation to test whether there is a difference between reported significant and non-significant correlations. Analysis revealed that the correlations between representational gestures and internal details for past and future conditions are not significantly different from each other (z = 1.20, p = .23). This additional analysis is now added under the “Results” section as well (p. 18). 

As for the multiple comparisons, in the original submission, we considered not to go forward with the corrections. The reasons are: First, the quality of this study is exploratory rather than completely confirmatory. Second, correlations are not that multiple. We would be dividing by 7 (for the most significant correlation) at most to conduct a Bonferroni-Holm correction. Finally, the tests we ran were preplanned. We did not run all the possible correlations available. For these reasons, by doing corrections, we may risk committing a Type II error. 

COMMENT 5:

Discussion: One issue to discuss is finding that past, future autobio and non-autobio events related to gesture use and then finding/discussing how details correlate the gesture use only for the autobiographical condition. The pairing of these findings suggest to me that the correlations reported could reflect a measure of on-topic content (ie., internal details for autobio events) relating to gesture use. Would there be a way to test this or perhaps speak to this in the discussion? Perhaps relation to broader embodied cognition work in the discussion as well.

RESPONSE: 

If the reviewer’s point was whether the gesture use was due to mapping with episodic content; i.e., “on-topic content, yes, this was indeed our point. Even in the non-autobiographical condition, individuals would draw on previous past experiences to some extent to come up with an abstraction (a typical X event), and therefore, an association with gesture is not surprising. We elaborated on this idea in the Discussion and refer to the embodied cognition literature. 

Reviewer #2: 

The manuscript describes a study investigating the relationship between gesture and autobiographical event (re)construction. By integrating standard research methods for assessing and coding for autobiographical event processing (narration of past, future and non-autobiographical events) and gesture (representational and non-representational) the authors have developed a novel and creative paradigm for further investigating features of embodied cognition. As predicted, the authors identified a relationship between the frequency of representational gestures and episodic features of autobiographical events. Namely, that representational gesture was associated with the frequency of internal details and reliving in past autobiographical events. In addition, relationships between representational and non-representational gestures and external details were observed for both past and future autobiographical events. Overall the topic area is novel and highlights potential new avenues of research into embodied cognition. The introduction provides a solid theoretical basis for conducting the research and clear potential hypotheses to be investigated. However, a number of points should be addressed to improve the clarity and understanding of this study prior to publication. I have outlined these points in more detail below.

Thank you very much for your overall encouraging evaluation.

COMMENT 1:

Calculation of participant sample size and presentation of associated analyses

The basis for the sample size selected for this study should be stated, it is currently unclear if the sample size is based on any a priori power analyses or previous literature in the field. This is of particular relevance in relation to the null findings when examining the first hypothesis comparing overall gesture rate between the three autobiographical conditions. It is relevant to determine if null findings could be a result of low power. If this is a possibility, it may be relevant to state this in the discussion. 

RESPONSE: 

We apologize for this omission. We have now added the justification for sample size under Participants on page 10. The inserted text is as follows:

‘Our sample size is determined by following the previous studies in the literature (e.g., Cook et al. 2010; Cook & Tanenhaus; 2009; Church et al., 2007; Ping et al., 2014). Additionally, post-hoc power analysis suggests that our sample of 41 people successfully detects the power of .85, with a false positive rate of .05 under the estimated effect size of the study (d = 0.44). This information was now added under Method → Participants on page 10.’

COMMENT 2:

Relatedly, the specific statistical analyses conducted to test this hypothesis, and the role of visuospatial skills and spatial imagery skills are not reported. There are also minor errors in the Note in Figure 1 (“None-ABM) and the graph legend replicates information already presented in the figure.

RESPONSE: 

The nonsignificant results for the visuospatial skills and spatial imagery skills are now reported in the results section. Figure 1 is also revised. We thank the reviewer for their detailed read.

COMMENT 3:

Coding of event and gesture narratives: Interrater reliability

Percentage agreement between raters was used as a measure of interrater reliability for the event narratives. This measure of reliability has been known to overestimate the level of agreement between raters. Inter-rater reliability statistics such as Cohens Kappa may provide a better estimate of interrater reliability. It is also unclear if interrater reliability was conducted for the gesture coding?

RESPONSE: 

This was an oversight. We thank the reviewer for pointing to it. We added kappa values for the gesture coding. The following text was added in the Method section under Gesture Coding (p. 13):

“Interrater reliability was computed by two independent raters coding 20% of the dataset. Cohen’s K (Cohen, 1960) was computed to be .796 (SD = 0.035) which indicates a moderate-to-strong agreement among the coders.” 

Regarding the calculation method of interrater reliability for the event narratives, we opted for the intraclass correlations (ICC) for two reasons: 

(1) It is the typical method for reporting the reliability in the Autobiographical Interview (AI) technique (for a recent study, please refer to Sheldon et al., 2020), and we would like to follow the standard in the literature.

(2) ICCs are commonly reported in the autobiographical memory literature mostly due to raters dealing with too many categories. In a complex coding scheme such as the AI with many categories, it is very hard for coders to simply guess and mistakenly seem like they are agreeing on a code. Intraclass correlation coefficients calculated for internal and external details were listed in the manuscript, under the “Event narrative coding” section (p. 11). For internal details, the intraclass correlation coefficient (ICC; one-way random effect model, McGraw & Wong, 1996) indicated excellent agreement with .99 and good agreement for external details with .88 ICC score.

COMMENT 4:

The structure and organization of the results

While the results are interesting, this section of the manuscript is very difficult to follow, as information is missing, reported several times and lacks clarity in places. It may be worthwhile to include an Analytical Plan section at the start of the results to present the rationale for the selection of analyses conducted to test the hypotheses outlined and any additional exploratory analyses. Presenting the correlations between event details and gesture in the form of a table may also improve the clarity of the results, and ensure that all relevant correlations are included.

RESPONSE: 

Having gone over the Results section, we agree that it is rather hard to follow. We thank the reviewer for pointing that out. As suggested, we resorted to include an Analytic Plan section at the beginning of the “Results” section (p. 13). We also summarize all the reported correlations in a newly added Tables 2 & 3(p. 19-20). 

COMMENT 5:

As outlined above, the type of statistical analyses conducted to test the first hypothesis are not stated explicitly. The statistics for a number of relevant correlations are also missing. For example, the correlation between representational gesture and external details and the non-significant correlations between gesture rate in future events and phenomenology (which are stated but the statistics are not reported). In contrast, other correlations, such as the correlation between representational details and external details in future events are reported twice. It is also unclear as to why the correlations between total number of gestures and event details are included and what additional value they provide.

RESPONSE: 

We included all analyses in the Analytic Plan section, and we now report the nonsignificant results. Also, we removed the correlations with the total number of gestures from the main text as the reviewer suggested. We thank the reviewer for a close read.

COMMENT 6:

Discussion

In the discussion the findings related to the relationship between representational gestures and episodic detail are clearly presented and the associated links to previous research and future implications described. However, it may also be relevant to discuss if the gesture-conceptualization framework may or may not account for the relationship between representational gestures and external details in terms of schematized self-knowledge which is observed in both past and future thinking in the present study (Kita et al. 2017).

RESPONSE: 

Our findings actually did not reveal an association between representational gestures and external details in the past condition. We, however, observed such an association in the future condition. Gesture-for-conceptualization framework proposes that gestures help schematizing information by way of assuming different functions, such as activating, manipulating, packaging and exploring information. Therefore, it is possible that a higher number of non-episodic elements, such as external details, to be present in the narratives with increased gesture use because external details are thought to reflect schematic information. However, since this was a posthoc explanation and was not confirmed in the past condition, we resort to add it as direction for future research on p. X. 

Minor points

P11, line 219 the term “atemporal” is used, it is not clear what this means in the context of the non-autobiographical event.

RESPONSE: 

We now changed the terminology to “procedural” following a similar study (Hilverman et al., 2016). The text now reads as: 

“In the non-autobiographical event condition, they were asked to narrate a procedural event (i.e., a typical bank transaction) to somebody who does not know anything about this world’s affairs.”

P11 lines 225-226 suggest that only the scales for vividness, reliving and mental time travel were utilized in the present study. However, in the results p 15, line 30, statistics related to emotional valence and intensity are reported. Were the measures reported in this manuscript (reliving, vividness and mental time travel) a subset of a larger number of items included in the study which also included valence and intensity?

These results on valence and intensity are not reported elsewhere in the paper so I was unsure of their relevance. It was interesting to note that valence negatively correlated with total number of gestures per utterance.

RESPONSE: 

Yes, as the reviewer suspected, measures for vividness, reliving and mental time travel (MTT) are a subset of a larger set of memory characteristics items (Johnson et al., 1988; Rubin et al., 2003; Butler et al., 2016). These larger set of memory characteristics items were included in the present design for exploratory purposes. We only had predictions regarding reliving, vividness and MTT because they constitute the associated subjective sense of remembering; i.e., autonoetic consciousness, which would give the episodicity sense to the narrated memories. Therefore, we only report those results. 

P7, line 128, the authors use the term “episodicness” and I wonder if the term episodicity may be more appropriate (see Habermas & Diel, 2013)

RESPONSE: 

We agree with the reviewer and have changed the terminology to episodicity. 

P19, line 374-376 the authors suggest that future events may resemble non-autobiographical events to a greater extent than anticipated. One way to examine the content of the events reported would be to code and compare the frequency of internal and external details in the events. A lower level of internal details in the non-autobiographical events may support the validity of the differentiating the autobiographical and non-autobiographical events in terms of autobiographical content.

RESPONSE: 

We agree with the reviewer that directly comparing autobiographical and non-autobiographical events in terms of internal details would be informative. However coding the memory details (i.e., internal and external details) is not possible, and also not theoretically justified for non-autobiographical events. The present coding scheme (The Autobiographical Interview; Levine et al. (2002) is based on selecting a main autobiographical event that happened at a specific time and place in the narrative. This would not be applicable to non-autobiographical events. As we discuss in the main text that while it is highly possible that knowledge of how a typical bank transaction event would unfold would be based on previous episodic experiences, non-autobiographical events are not tied to particular time and space. In this sense, they are closer to semantic memory than episodic memory. 

Reviewer #3: 

This is the first study I know of that looks at gesture use in autobiographical memory in adults. So many previous studies on gestures have focused on cartoon retelling tasks, this is a breath of fresh air! It was an interesting approach to look at the episodic specificity as a predictor of gesture production. The null findings for the cognitive abilities predicting individual differences also adds to the literature.

Thank you very much for your overall encouraging evaluation.

COMMENT 1:

The framing of the article could be tightened on two grounds: 1) how the genre might make a difference and 2) how different gesture types might matter. The introduction does get to the point that it might be the visuospatial content of speech that predicts gesture use. The introduction could get there faster. As for gesture type, it is not entirely clear why non-representational gestures were included in the analyses (yes, the results were intriguing, but it is not clear how to interpret them!).

RESPONSE: 

We have now added parts in the main text to better align with the first point made by the reviewer. Also based on other reviewers’ inquiries, we highlighted the link between autobiographical memory and gesture use in both the Introduction and Discussion sections, and stressed potential connections to the embodiment literature as far as the present data allows for (please see the highlighted sections in the revised manuscript).

As for the second point, after deliberation, we decided to keep non-representational gestures in the analyses due to the fact that this study is the first of its kind; in linking episodic processes and gesture use. Even though we did not have framework-guided predictions regarding non-representational gestures, we felt it is important to share the data fully. Even though there is less consensus on the functional role of non-representational gestures, they are frequently discussed within the discourse-fluency contexts (Stitzlein et al., 2005; Vila-Gimenez & Prieto, 2020), we tentatively expect them to be correlated with the external details because evidence from the autobiographical literature also suggests that the reason external details, and therefore, semantic information, are included in the autobiographical narratives is to elaborate or “embellish” the discourse (Devitt et al., 2017, p. 1). It is not surprising, then, when non-representational gestures assume a fluency resolving role, they would enhance the generation of semantic information in autobiographical narratives. After adding that this is a tentative, and post hoc interpretation in the revised manuscript, we, then, suggest future directions. (p.25-26)

COMMENT 2:

Given the wide individual variability in gesture rate reflect an individual tendency to gesture a lot (or a little)? In other words, were there correlations in gesture rate across conditions? If so, that would support the argument that gesturing reflects (at least in part) an individual’s characteristics. Even if no support for that individual aspect being visuospatial ability was found in this study.

RESPONSE: 

This is a valuable point raising a concern about individual differences. The correlations in gesture rate across conditions were added on p.16 under the“Results” section. There were indeed correlations between past and future conditions which would indicate general tendencies for gesturing. We also added this finding and discussed (p. 16). 

Somewhat smaller points:

-There is no exposition of the linguistic construction that was used as the baseline (either words or utterances). I think it was words, but were false starts and self-repetitions counted? Why or why not? And words were orthographic words? If utterances, then what was the definition of “utterance”?

RESPONSE:

Thank you for pointing out these crucial aspects of the process of transcription. At the event narrative coding section (p. 11 ), we explained the definition of utterances and indicated that all narratives were coded verbatim as you can see below. 

‘Memory narratives were transcribed verbatim from the video recordings with all utterances 

- I don’t understand why participants who did not gesture at all were not included in the analyses. At least one study (with children) found that not gesturing led to reduced visuospatial content (Laurent et al., 2020).

RESPONSE:

Our original reasoning was that not including individuals who did not gesture would make the conclusions stronger (i.e., gestures lead to episodic content). This had been discussed quite extensively within the research team. We ended up including the participants who did not gesture. However, much to our embarrassment, we forgot to remove the portion of the text that described our reasons for not including them. Thus, the numbers in the first submission actually reflect the full set of participants, including the ones who did not gesture. We now remove that bit of the text in the revised version.

For comparison reasons, and for the reviewer’s attention, we re-ran the analyses without those participants who did not gesture during the narrative. Item A below shows those results, and item B shows the full set. As you can see the numbers only slightly differ from each other.

A. For the past events, representational gestures correlated with internal details (τ = .33, p = .005). To delve into further examination of which particular episodic details were related to the representational gestures, we looked into internal detail categories. Representational gestures were correlated with internal place details (τ = .30, p = .02) as well as internal perceptual details (τ = .38, p = .002), event details (τ = .23, p = .048).

B. For the past events, representational gestures correlated with internal details (τ = .27, p =.02). To delve into further examination of which episodic details were related to the representational gestures, we looked into internal detail categories. Representational gestures were correlated with internal place details (τ = .28, p = .02) as well as internal 

-It could make the results easier to follow to include a table with all the correlations.

RESPONSE: 

In the revised version of the manuscript we add a table summarizing the described correlations 

-Were any corrections be made for multiple correlations?

RESPONSE: 

After some deliberation, we decided not to do corrections for three reasons: First, this was an exploratory study, and second, we only conducted some pre-planned analyses instead of trying all possible correlations. Finally, correlations were not that “multiple”. When applying a Bonferroni-Holm correction we had to divide by 7 for the most significant correlation. According to many sources, in these cases, one would increase the risk of committing Type II errors which 

-How was the subjective sense of recollection measured?

RESPONSE: 

Subjective sense of recollection is measured for each memory by using a selection of items from the standard scales in the literature, namely Memory Characteristics Questionnaire (Johnson et al., 1988) and the Autobiographical Memory Questionnaire (Rubin et al., 2003; Butler et al., 2016). They were selected based on the previously reported relations between individual differences in visual imagery and phenomenological experience as well as their applicability for both past and future events (e.g., D’Argembeau & Van der Linden, 2006; Ogden & Barker, 2001; Vannucci et al., 2016). For instance, a typical item for measuring the sense of reliving would ask the participants to rate the following statement on a 5-point Likert scale: “While remembering the event, I feel as if I am reliving it.” These items were distributed as paper-pencil questionnaires. These are described in the manuscript under the “Materials and procedure” section (p. 11). 

-I don’t know if it helps at all, but I know of at least one study that included autobiographical memories with children (Marentette et al., 2020). It might not be useful since the focus of that study was different. That said, if I remember correctly, the gesture rate was lower when the children told autobiographical stories than when they told fictional stories.

RESPONSE: 

We thank the reviewer for highlighting this article that we missed. We now cite this article in the “Introduction” section, where we strengthen the previous literature that investigates the link 

Very small points:

p. 3, second paragraph, line 2: what makes a ‘type’ of gesture?

RESPONSE: 

We meant “gesture categories” and we now changed the sentence. 

p. 5, starting line 4: It was unclear whether the participants in Cook et al. (2010) gestured or if the events themselves were highly visuospatially imagistic

RESPONSE: 

We also believe this is an important point and now expanded this aspect in more detail in the corresponding paragraph (p. 5). The reported relationship between gesture use and memory retrieval were present for both high and low visuospatial stimuli that were tested in the same study. We now highlight these findings in the Introduction.

p. 5, sentence starting on line 4 from the bottom: later on in the paper, the authors make the argument that the directionality might be bidirectional.

RESPONSE: 

We thank the reviewer for pointing out an important point. We revised the text to better align with relevant work. Specifically, our initial prediction that the relationship between memory and gesture use might bi bidirectional following the previous literature (Alibali et al., 2014). However, the directionality account, which is hippocampal representations supports gesture use, suggested by the Hilverman et al. (2016) study is emphasized in detail in the “Discussion” section (p. 23).

p. 12, last sentence: how were these percentages calculated? Particularly, what was in the denominator?

RESPONSE: 

The denominator in this case was the total number of details (internal or external). Percentages were calculated by dividing the total number details in the relevant category by the number of agreements. That proportion was then converted to a percentage. However, we now report event narrative interrater agreement as intraclass correlation coefficient (ICC) as can be seen on page 14. 

ICC is computed by the following formula: 

ICC = (variance of interest) / (total variance) = (variance of interest) / (variance of interest + unwanted variance) 

p. 13, first line of text: what does “representative” mean? Were self-adaptors included if they were rhythmic?

RESPONSE: 

“Representative” is gestures depicting particular actions, objects, abstract concepts and/or their properties. We now extended this sentence by providing examples of what we mean by gesture being “representative” (p. 14). We also added that our beat gestures did not include self-adaptors such as self-touch or bending fingers. 

p. 13: note that the description here is in terms of gestures per words while the Figure shows gestures per utterance.

RESPONSE: 

We now revised the corresponding parts (i.e., “Results” section) as “gesture per utterances” to provide internal consistency for the terminology used. 

p. 13, last line: explain what the post-hoc analysis was exactly.

RESPONSE: 

We included the type of the post-hoc analysis to the relevant sentence, which was Tukey in this particular case.

p. 16, midway through the first paragraph about the correlation between representational gestures and internal details. Was this correlation with the NUMBER of gestures or the gesture RATE? If the former, couldn’t this correlation come about because people who talked more gestured more?

RESPONSE: 

Although we calculated gesture rates across gesture types in some analyses (e.g., mean gesture rates across gesture types), we used the number of gestures in each gesture category when conducting correlational analyses. That is because, when we take the gesture rates to normalize gesture use in a given narrative, we also need to normalize the number of event details for narrative length. In that case, we have narrative length (i.e., total number of spoken utterances and words) at the denominators of both variables and in a correlational analysis they cancel out each other. Therefore, even though we did not use gesture rates in some analyses, we considered narrative length. The reason to use number of gestures was that our episodic specificity measure (the AI) uses count information as standard procedure. We, therefore, wanted to follow common practice in the literature.

p. 16, for the correlations that the authors would really like to remain salient in readers’ minds, scatterplots could be useful.

RESPONSE: 

Yes, for that purpose, we include two tables summarizing the correlations. Please refer to Tables 2 & 3 on pages 19-120. Scatterplots would be too many and complicated for readers to follow, we thought.

p. 19, first paragraph: it would be useful if the authors reported the number of details across conditions in the results section so readers can follow their arguments here.

RESPONSE: 

These data are summarized in Table 1 (p. 17). We included several references to this table in the “Results” and “Discussion” sections where relevant.

p. 22, last sentence: this point was already made

RESPONSE: 

The earlier reference to this point was removed. Thank you.

References

Arslan, B., & Göksun, T. (2021). Aging, working memory, and mental imagery: Understanding gestural communication in younger and older adults. Quarterly Journal of Experimental Psychology, 74(1), 29-44. 

Arslan, B., & Göksun, T. (2022). Aging, gesture production, and disfluency in speech: A comparison of younger and older adults. Cognitive Science, 46(2), e13098. 

Butler, A. C., Rice, H. J., Wooldridge, C. L., & Rubin, D. C. (2016). Visual imagery in autobiographical memory: The role of repeated retrieval in shifting perspective. Consciousness and cognition, 42, 237-253.

Church, R. B., Garber, P., & Rogalski, K. (2007). The role of gesture in memory and social communication. Gesture, 7(2), 137-158.

Cook, S. W., Yip, T. K., & Goldin-Meadow, S. (2010). Gesturing makes memories that last. Journal of memory and language, 63(4), 465-475.

Cook, S. W., & Tanenhaus, M. K. (2009). Embodied communication: Speakers’ gestures affect listeners’ actions. Cognition, 113(1), 98-104.

D’Argembeau, A., & Van der Linden, M. (2006). Individual differences in the phenomenology of mental time travel: The effect of vivid visual imagery and emotion regulation strategies. Consciousness and cognition, 15(2), 342-350.

Devitt, A. L., Addis, D. R., & Schacter, D. L. (2017). Episodic and semantic content of memory and imagination: A multilevel analysis. Memory & cognition, 45(7), 1078-1094.

Hilverman, C., Cook, S. W., & Duff, M. C. (2016). Hippocampal declarative memory supports gesture production: Evidence from amnesia. Cortex, 85, 25-36.

Johnson, M. K., Foley, M. A., Suengas, A. G., & Raye, C. L. (1988). Phenomenal characteristics of memories for perceived and imagined autobiographical events. Journal of Experimental Psychology: General, 117(4), 371. 

Levine, B., Svoboda, E., Hay, J. F., Winocur, G., & Moscovitch, M. (2002). Aging and autobiographical memory: dissociating episodic from semantic retrieval. Psychology and aging, 17(4), 677.

McGraw, K. O., & Wong, S. P. (1996). Forming inferences about some intraclass correlation coefficients. Psychological methods, 1(1), 30.

Ogden, J. A., & Barker, K. (2001). Imagery used in autobiograpical recall in early and late blind adults. Journal of Mental Imagery-New York International Imagery Association, 25(3/4), 153-176.

Ping, R. M., Goldin-Meadow, S., & Beilock, S. L. (2014). Understanding gesture: Is the listener’s motor system involved?. Journal of Experimental Psychology: General, 143(1), 195.

Rubin, D. C., Schrauf, R. W., & Greenberg, D. L. (2003). Belief and recollection of autobiographical memories. Memory & cognition, 31(6), 887-901.

Sheldon, S., Williams, K., Harrington, S., & Otto, A. R. (2020). Emotional cue effects on accessing and elaborating upon autobiographical memories. Cognition, 198, 104217.

Stitzlein, C. A., Trafton, J. G., & Trickett, S. B. (2005). Simple Gesture Analysis in Narrative Speech: Expert-novice Differences. In Proceedings of the Annual Meeting of the Cognitive Science Society (Vol. 27, No. 27).

Vannucci, M., Pelagatti, C., Chiorri, C., & Mazzoni, G. (2016). Visual object imagery and autobiographical memory: Object Imagers are better at remembering their personal past. Memory, 24(4), 455-470.

Vilà-Giménez, I., & Prieto, P. (2020). Encouraging kids to beat: Children's beat gesture production boosts their narrative performance. Developmental Science, 23(6), e12967.

---

## [Decision Letter · Decision Letter 1]

13 Jan 2023

PONE-D-22-18508R1The role of gestures in autobiographical memoryPLOS ONE

Dear Dr.Aydin,,

Thank you for submitting your manuscript to PLOS ONE. After careful consideration, we feel that it has merit but does not fully meet PLOS ONE’s publication criteria as it currently stands. Therefore, we invite you to submit a revised version of the manuscript that addresses the points raised during the review process.

I have received reviews from two of the reviewers who read your paper previously. Both reviewers and myself agree that the manuscript is significantly improved. One reviewer has recommended publication while the second reviewer requires further minor changes. I agree with the suggestions of the second reviewer. Many changes concern defining terms or labelling Figures or Tables more clearly. There are also a few points about how you analysed the data. You should address all points raised or justify why you are not addressing the point.. 

Please submit your revised manuscript by Feb 27 2023 11:59PM. You can also submit earlier than that date. If you will need more time than this to complete your revisions, please reply to this message or contact the journal office at plosone@plos.org. Please include the following items when submitting your revised manuscript:A rebuttal letter that responds to each point raised by the academic editor and reviewer(s). You should upload this letter as a separate file labeled 'Response to Reviewers'.A marked-up copy of your manuscript that highlights changes made to the original version. You should upload this as a separate file labeled 'Revised Manuscript with Track Changes'.An unmarked version of your revised paper without tracked changes. You should upload this as a separate file labeled 'Manuscript'.If applicable, we recommend that you deposit your laboratory protocols in protocols.io to enhance the reproducibility of your results. Protocols.io assigns your protocol its own identifier (DOI) so that it can be cited independently in the future. For instructions see: https://journals.plos.org/plosone/s/submission-guidelines#loc-laboratory-protocols. Additionally, PLOS ONE offers an option for publishing peer-reviewed Lab Protocol articles, which describe protocols hosted on protocols.io. Read more information on sharing protocols at https://plos.org/protocols?utm_medium=editorial-email&utm_source=authorletters&utm_campaign=protocols.

We look forward to receiving your revised manuscript.

Kind regards,

Barbara Dritschel, PhD

Academic Editor

PLOS ONE

Journal Requirements:

Reviewers' comments:

Reviewer's Responses to Questions

**Comments to the Author**

1. If the authors have adequately addressed your comments raised in a previous round of review and you feel that this manuscript is now acceptable for publication, you may indicate that here to bypass the “Comments to the Author” section, enter your conflict of interest statement in the “Confidential to Editor” section, and submit your "Accept" recommendation.

Reviewer #1: All comments have been addressed

Reviewer #3: All comments have been addressed

2. Is the manuscript technically sound, and do the data support the conclusions?

Reviewer #1: Yes

Reviewer #3: Yes

3. Has the statistical analysis been performed appropriately and rigorously? 

Reviewer #1: Yes

Reviewer #3: Yes

4. Have the authors made all data underlying the findings in their manuscript fully available?

Reviewer #1: Yes

Reviewer #3: Yes

5. Is the manuscript presented in an intelligible fashion and written in standard English?

Reviewer #1: Yes

Reviewer #3: Yes

6. Review Comments to the Author

Reviewer #1: I believe the authors did a great job addressing prior concerns. I have no other comments with the paper.

Reviewer #3: This manuscript reads quite well and can add to the literature on the function of gestures.

I have a few minor points that the authors can easily address:

p. 3, paragraph 2, penultimate sentence: it's not clear from the wording how deictic gestures are classified; reword for clarity.

p. 10, fifth line from the bottom: could you briefly characterize what a "flag ceremony" is? If these were American participants, it would probably mean swearing allegiance. But I suspect that this flag ceremony involves some different activities?

p. 13, line 3 from the bottom: how were word tokens calculated? Notably were false starts and self-repetitions included or not? (I don't care if they were or not; I just want the specification so that future researchers can do the same!)

p. 14, Figure 1: the label of the x-axis says "gestures per utterance" but the text says that it was gestures per word that was calculated. If it was utterances, make sure that "utterance" is defined.

p. 14, first line of text: exactly what analysis was done?

p. 14, penultimate line: I will simply register my disagreement here with the authors' decision to throw out the zeros. If they think that gesturing leads to the inclusion of more episodic details, then not gesturing at all should mean the inclusion of few episodic details.

p. 14, last line, "utterance": utterance or word??????

p. 15, Table 1: consider putting # or N in front of the gesture lines to make it clear that this is not the gesture rate (i.e., per word or per utterance, whichever way it was actually done!).

p. 21, end of first paragraph: here I was wondering whether the representational gesture rate and non-representational gesture rate was correlated within a task? It might just be that it is the 'gesturiness' of a person that predicts the episodic details...

7. PLOS authors have the option to publish the peer review history of their article (what does this mean?). If published, this will include your full peer review and any attached files.

Reviewer #1: No

Reviewer #3: No

---

## [Author Response · Author response to Decision Letter 1]

16 Jan 2023

We thank you for this opportunity to improve on the quality of our manuscript based on the rich set of comments provided by the Editor and the Reviewers. Below, we provide detailed point-by-point responses (italicized) to these comments.

p. 3, paragraph 2, penultimate sentence: it's not clear from the wording how deictic gestures are classified; reword for clarity.

RESPONSE: Thank you. The text is changed. Please see below:

“deictic gestures include the type of hand movements that involve pointing or reaching to the objects or abstract entities (McNeill, 1992).”

p. 10, fifth line from the bottom: could you briefly characterize what a "flag ceremony" is? If these were American participants, it would probably mean swearing allegiance. But I suspect that this flag ceremony involves some different activities?

RESPONSE: We changed the name to “Flag-raising morning ceremony”. We added a note explaining what it entails. Please see below:

“... a typical flag raising morning ceremony in school, which is a common practice in some countries where students gather in the school yard to sing the national anthem together.”

p. 13, line 3 from the bottom: how were word tokens calculated? Notably were false starts and self-repetitions included or not? (I don't care if they were or not; I just want the specification so that future researchers can do the same!)

RESPONSE: The text below was added to the main text.

“Memory narratives were transcribed verbatim from the video recordings with all utterances including false starts, fillers, and other disfluencies in addition to words.” 

p. 14, Figure 1: the label of the x-axis says "gestures per utterance" but the text says that it was gestures per word that was calculated. If it was utterances, make sure that "utterance" is defined.

RESPONSE: This is fixed as utterance throughout the text where relevant. In page 12, we note that “....with all utterances including false starts, fillers, and other disfluencies in addition to words.”

p. 14, first line of text: exactly what analysis was done?

RESPONSE: Kendall’s Tau rank correlation analysis. This is noted in the Analytical Plan section.

p. 14, penultimate line: I will simply register my disagreement here with the authors' decision to throw out the zeros. If they think that gesturing leads to the inclusion of more episodic details, then not gesturing at all should mean the inclusion of few episodic details.

RESPONSE: The short answer to this comment is: Yes, we agree with the Reviewer and did not throw out the zeros. Please see the longer response below:

“Our original reasoning was that not including individuals who did not gesture would make the conclusions stronger (i.e., gestures lead to episodic content). This had been discussed quite extensively within the research team. We ended up including the participants who did not gesture. However, much to our embarrassment, we forgot to remove the portion of the text that described our reasons for not including them. Thus, the numbers in the first submission actually reflect the full set of participants, including the ones who did not gesture. We now remove that bit of the text in the revised version.

For comparison reasons, and for the reviewer’s attention, we re-ran the analyses without those participants who did not gesture during the narrative. Item A below shows those results, and item B shows the full set. As you can see the numbers only slightly differ from each other.

A. For the past events, representational gestures correlated with internal details (τ = .33, p = .005). To delve into further examination of which particular episodic details were related to the representational gestures, we looked into internal detail categories. Representational gestures were correlated with internal place details (τ = .30, p = .02) as well as internal perceptual details (τ = .38, p = .002), event details (τ = .23, p = .048).

B. For the past events, representational gestures correlated with internal details (τ = .27, p =.02). To delve into further examination of which episodic details were related to the representational gestures, we looked into internal detail categories. Representational gestures were correlated with internal place details (τ = .28, p = .02) as well as internal perceptual details (τ = .36, p = .002) as depicted in Table 2.

p. 14, last line, "utterance": utterance or word??????

RESPONSE: It should be “utterance”. 

p. 15, Table 1: consider putting # or N in front of the gesture lines to make it clear that this is not the gesture rate (i.e., per word or per utterance, whichever way it was actually done!).

RESPONSE: We added a note under Table 1 to indicate that the values are, in fact, counts. 

p. 21, end of first paragraph: here I was wondering whether the representational gesture rate and non-representational gesture rate was correlated within a task? It might just be that it is the 'gesturiness' of a person that predicts the episodic details…

RESPONSE: Yes, as presented in Tables 2 & 3, they are correlated. However, it was only in the past event condition, we were able to spot a relationship between episodic details and representational gestures. We did not observe non-representational gestures to be associated with episodic details in any condition.

Interestingly though, non-representational gesture use was also correlated with the subjective sense of recollection. Although prior work has not tested this relationship, the evidence on gestures alleviating cognitive load in narrative recall and problem-solving (e.g., Cook et al., 2010; Lin, 2021; Overoye & Wilson, 2020; Ping & Golden-Meadow, 2010) suggests that reduced cognitive demands during retrieval via the use of gestures may lead to richer reflective recollective experience. This is added in the Discussion.

Additional changes: 

Although it was not raised by the reviewers and the editor, we realized mismatches when reporting the statistics in the Results sections. Thus, we revised this section by italicizing the "p" denotation via track-changes.

We also updated the references section for the added and removed citations in the revised version of the paper.

---

## [Editor Report · Decision Letter 2]

27 Jan 2023

PONE-D-22-18508R2The role of gestures in autobiographical memoryPLOS ONE

Dear Dr. Aydin,

Thank you for submitting your manuscript to PLOS ONE. After careful consideration, we feel that it has merit but does not fully meet PLOS ONE’s publication criteria as it currently stands. Therefore, we invite you to submit a revised version of the manuscript that addresses the points raised during the review process.

You have done an excellent job on this second revision and addressed all points raised. There are problems with spacing ( gaps were there should be none) and a spelling error on p. 9 indeces should be indices. Can you carefully check these points and resubmit. 

Please submit your revised manuscript by 6/02/2023. If you will need more time than this to complete your revisions, please reply to this message or contact the journal office at plosone@plos.org. Please include the following items when submitting your revised manuscript:A rebuttal letter that responds to each point raised by the academic editor and reviewer(s). You should upload this letter as a separate file labeled 'Response to Reviewers'.A marked-up copy of your manuscript that highlights changes made to the original version. You should upload this as a separate file labeled 'Revised Manuscript with Track Changes'.An unmarked version of your revised paper without tracked changes. You should upload this as a separate file labeled 'Manuscript'.If applicable, we recommend that you deposit your laboratory protocols in protocols.io to enhance the reproducibility of your results. Protocols.io assigns your protocol its own identifier (DOI) so that it can be cited independently in the future. For instructions see: https://journals.plos.org/plosone/s/submission-guidelines#loc-laboratory-protocols. Additionally, PLOS ONE offers an option for publishing peer-reviewed Lab Protocol articles, which describe protocols hosted on protocols.io. Read more information on sharing protocols at https://plos.org/protocols?utm_medium=editorial-email&utm_source=authorletters&utm_campaign=protocols.

We look forward to receiving your revised manuscript.

Kind regards,

Barbara Dritschel, PhD

Academic Editor

PLOS ONE
---

## [Author Response · Author response to Decision Letter 2]

30 Jan 2023

Dear Editor,

We thank you for this opportunity to improve on the quality of our manuscript based on the rich set of comments provided by the Editor and the Reviewers. Below, we provide detailed point-by-point responses (italicized) to these comments.

Editor: “There are problems with spacing ( gaps were there should be none) and a spelling error on p. 9 indeces should be indices. Can you carefully check these points and resubmit.”

RESPONSE: Thank you. We have addressed all of these points in the submitted revised version.

---

## [Editor Report · Decision Letter 3]

1 Feb 2023

The role of gestures in autobiographical memory

PONE-D-22-18508R3

Dear Dr. Aydin, 

We’re pleased to inform you that your manuscript has been judged scientifically suitable for publication and will be formally accepted for publication once it meets all outstanding technical requirements.

Kind regards,

Barbara Dritschel, PhD

Academic Editor

PLOS ONE
---

## [Editor Report · Acceptance letter]

14 Feb 2023

PONE-D-22-18508R3 

The role of gestures in autobiographical memory 

Dear Dr. Aydin:

I'm pleased to inform you that your manuscript has been deemed suitable for publication in PLOS ONE. Congratulations! Your manuscript is now with our production department. 

Kind regards, 

on behalf of

Dr. Barbara Dritschel 

Academic Editor

PLOS ONE